

# Incorporating $^{15}$N into the outputs of SMOKE version 4.6 as the emission
input dataset for CMAQ version 5.2.1 for assessing the role emission
sources plays in controlling the isotopic composition of NO$_x$, NO$_y$, and
atmospheric nitrate
*Huan Fang[†] and Greg Michalski[†‡]*
[†]Department of Earth, Atmospheric, and Planetary Sciences Purdue University, 550 Stadium Mall
Drive, West Lafayette, IN 47907, United States
[‡]Department of Chemistry, Purdue University, 560 Oval Drive, West Lafayette, IN 47907, United
States
Correspondence: Huan Fang, fang63@purdue.edu



Abstract
Nitrogen oxides ($NO_x$ = nitric oxide (NO) + nitrogen dioxides ($NO_2$)) are important trace gases
that affect atmospheric chemistry, air quality, and climate. Contemporary development of $NO_x$
emissions inventories is limited by the understanding of the roles of vegetation (net $NO_x$ source or
net sink), gasoline and diesel in vehicle emissions, and application of $NO_x$ emission control
technologies. In this study, we used the nitrogen stable isotope composition of $NO_x$ ($\delta^{15}N(NO_x)$)
to resolve the uncertainties in $NO_x$ emission sources, by incorporating $^{15}N$ into the US EPA trace
gas emission model SMOKE (Sparse Matrix Operator Kernel Emissions) and compared simulated
spatiotemporal patterns in $NO_x$ isotopic composition with corresponding atmospheric
measurements in West Lafayette, Indiana, USA. The results indicate the potential underestimation
of emissions from soil, livestock waste, off-road vehicles, and natural gas power plants and the
potential overestimation of emissions from on-road vehicles and coal-fired power plants.

1.  Introduction

$NO_x$ are important trace gases that affect atmospheric chemistry, air quality, and climate ($NO_x$
= nitric oxide (NO) + nitrogen dioxide ($NO_2$)). The main sources of tropospheric $NO_x$ are
emissions from vehicles, power plants, agriculture, livestock waste, as well as the natural by-
product of nitrification and denitrification occurring in soil, and lightning. The $NO_x$ photochemical
cycle generates OH and $HO_2$ radicals, organic peroxy radicals ($RO_2$), and ozone ($O_3$), which
ultimately oxidize $NO_x$ into $NO_y$ ($NO_y$ = $NO_x$ + HONO + $HNO_3$ + $HNO_4$ + $N_2O_5$ + other N oxides).
During the photochemical processes that convert $NO_x$ to $NO_y$, ground-level concentrations of $O_3$
become elevated and secondary particles are generated. Secondary aerosols in turn affect cloud
physics, enhancing the reflection of solar radiation (Schwartz, 1996) and are hazardous to human
health (Lighty et al., 2000). Thus, the importance of $NO_x$ in air quality, climate, and human and
environmental health makes understanding the spatial and temporal variation in the sources of $NO_x$
a vital scientific question. However, despite years of research, there are still a number of significant
uncertainties in the $NO_x$ budget.
There are significant uncertainties in the amount of $NO_x$ emitted by soil at local and global
scales. About 15% of global $NO_x$ emissions, ranging from 6.6 to 21 Tg N yr$^{-1}$, is derived from
global soil $NO_x$ emissions yet evaluating and verifying emission rates using both laboratory and
field measurements is still a challenge (Galbally & Roy, 1978; Muller, 1992; Potter et al., 1996;
Yienger and Levy, 1995; Davidson and Kingerlee, 1997; Ganzeveld et al., 2002; Jaeglé et al., 2005;
Yan et al., 2005; Stehfest and Bouwman, 2006; Hudman et al., 2012). Soil $NO_x$ emissions vary by
different biome types, meteorological conditions, and soil physicochemical properties. Soil $NO_x$
emissions also depend on soil moisture that is a function of climate, such as in Mediterranean
climates and tropical savannas, where wet and dry seasons cause extreme fluctuations in soil
moisture (Davidson, 1992; Yienger and Levy, 1995; Scholes et al., 1997; Zörner et al., 2016). The
application of N fertilizer also has a strong effect on soil $NO_x$ emissions, which can dramatically
increase during the first 1-2 days after N fertilizer application and can take several weeks for the
emission rate to drop to pre-fertilizer levels (Ludwig et al., 2001). N fertilizers nitrogen may have
increased soil $NO_x$ emissions by up to 11% (Shepherd, 1991; Pilegaard, 2013), and probably
currently contributes 1.8 Tg N yr$^{-1}$ (Hudman, 2012). Furthermore, soil $NO_x$ emissions are likely to
increase as the worldwide use of fertilizers grows (Galloway et al., 2004; Houlton et al., 2013).
There is also a controversy about the fate of $NO_x$ emitted by the soil in terms of the amount that
escapes the canopy and mixes into the boundary layer. Previous research has highlighted the role



of vegetation in $NO_x$ removal, when the ambient $NO_x$ concentrations are below the "compensation
point" (i.e. between 1 and 3 ppbv), vegetation acts as a net source of atmospheric $NO_x$, but above
4 ppbv acts as a net sink (Johansson, 1987; Thoene, Rennenberg & Weber, 1996; Slovik et al.,
1996; Webber & Rennenberg, 1996). However, other research claims the up to 75% of soil $NO_x$
is lost through vegetation canopy reduction even when the ambient $NO_x$ concentration was as low
as 0.2 to 0.4 ppbv (Jacob & Wofsy, 1990; Hanson & Lindberg, 1991; Yienger & Levy II, 1995).
For example, soil $NO_x$ emission in California may be underestimated by up to 50% net due to the
sink by vegetation, significantly changing current the $NO_x$ emission inventory (Almaraz et al.,
9    2018).
10       On-road vehicles are one of the major sources of $NO_x$, yet there are also questions about
whether emission inventories are accurate. According to Parrish (2006), the estimation of on-road
vehicle $NO_x$ emission has at least 10 to 15% uncertainty. The algorithm used in the National
Emission Inventory (NEI), is mileage-based, which estimates $NO_x$ emission from on-road vehicles
by travel time, speed of travel on different roadways, and emissions from vehicles per distance
traveled. The emission factor of each vehicle classification and emission types are based on the
represented measurement of $NO_x$ from on-road vehicles in the US, under different ambient
temperatures, travel speeds, operating modes, fuel volatility, and mileage accrual rates (Dreher &
Harley, 1998; USEPA, 2003). However, the emission factors of vehicle classifications and
emission types are derived from the measurements at a relatively small number of sites. As a result,
the estimations of $NO_x$ emission from on-road vehicles by mileage-based approach appears to be
inconsistent with some on-road and ambient air measurements (Ingalls, 1989; Pierson et al., 1990;
Fujita et al., 1992; Pierson et al., 1996; Singer and Harley, 1996). For example, $NO_x$ emissions
from diesel engines are likely underestimated by a factor of 2 (Pierson et al., 1996; Cicero-
Fernandez et al., 1997; Sawyer et al., 2000) and estimates by the mileage-based approach does not
follow the same spatial and temporal patterns as the $NO_x$ measurements (Dreher & Harley, 1998).
An alternative is a fuel-based approach, which directly uses to estimates fuel consumption based
on gas tax data and derives the $NO_x$ emission by the emission factors in gram per gallon based on
the represented on-road measurements (Singer & Harley, 1996; Dreher & Harley, 1998). By doing
so, the only uncertainties are fuel sales data and emission factors, which are easier to determine
and get controlled. As a result, the emission inventories derived from the fuel-based approach are
closer to the measurements (Singer & Harley, 1996; Dreher & Harley, 1998; Sawyer et al., 2000;
Parrish, 2006). At the same time, however, the fuel-based approach fails to provide accurate spatial
or temporal $NO_x$ emissions (Sawyer et al., 2000).
34       The uncertainty in power plant $NO_x$ emissions is mainly the result of the recent
implementation of $NO_x$ emission control technologies. The Clean Air Act of 1995 required $NO_x$
emission control technologies to be implemented on new power plants. The major emission control
technologies are a). LNB: low $NO_x$ burner, which decreases $NO_x$ emission by lowering the oxygen
to nitrogen in the fuel; b). SCR: selective catalytic reduction, which chemically reduces $NO_x$ to $N_2$
by using $NH_3$ or urea as a reductant over a metal catalyst; c). SNCR: selective non-catalytic
reduction, converts $NO_x$ to $N_2$ by reacting $NO_x$ with $NH_3$ or urea; and d). OFA: over-fire air, which
increases the fuel combustion efficiency by introducing air during the combustion (Felix et al.,
2012; Srivastava et al., 2005; Xing et al., 2013). Between 1990 and 2010, In the United States,
$NO_x$ control technology used in coal-fired power plants increased from less than 20% to about
86%, and from less than 2% to 70% for natural gas power plants, which decreased overall US
power plant $NO_x$ emissions by about 70% (Xing et al., 2013). The reduction of $NO_x$ emission from
power plants varies by the facility, due to the choice of emission control technologies, which cause





the uncertainties. The removal efficiencies of $NO_x$ emission are also different for each control technology. LNB can remove up to 50% of $NO_x$ emissions from power plants but using LNB and OFA at the same time could remove 60% to 75%. SNCR can remove 30% to 66% while SCR can remove 80% to more than 90% of power plant $NO_x$ while reburning can remove 39% to 67% (Srivastava et al., 2005). All of these removal percentages, however, do not apply to initial fire-up times prior to catalyst efficiency reaching its maximum.

The nitrogen stable isotope composition of $NO_x$ might be a useful tool to help resolve the uncertainties of how $NO_x$ emission sources vary in space and time. Previous studies have shown that natural and anthropogenic $NO_x$ sources have distinctive $^{15}N/^{14}N$ ratios (Ammann et al., 1999; Felix et al., 2012; Felix and Elliott, 2013; Fibiger et al., 2014; Heaton, 1987; Hoering, 1957; Miller et al., 2017; Walters et al., 2015a, 2015b, 2018). This variability in $NO_x$ $^{15}N/^{14}N$ ratios quantified by

$$\delta^{15}N(NO_x) \text{ (‰)} = [(^{15}NO_x/^{14}NO_x) / (^{15}N_2/^{14}N_2)_{air} -1] \times 1000) \qquad \text{Eq. (1)}$$

where $^{15}NO_x/^{14}NO_x$ is the measurement of relative abundance of $^{15}N$ to $^{14}N$ in atmospheric $NO_x$, compared with the ratio of nitrogen in the air, which has a $^{15}N_2/^{14}N_2 = 0.0036$.

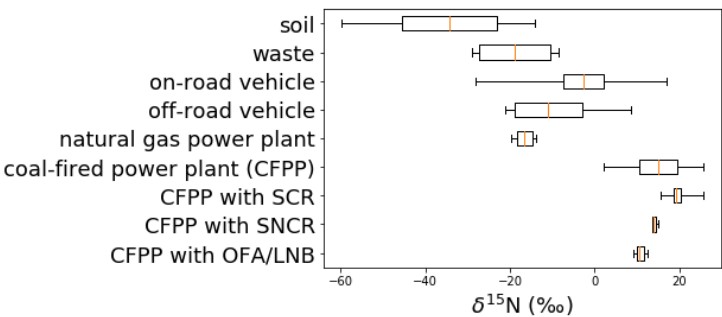

Figure 1: Box (lower quartile, median, upper quartile) and whisker (lower extreme, upper extreme) plot of the distribution of $\delta^{15}N$ values for various $NO_x$ emission sources.

Previous research has shown that there are distinctive differences in $\delta^{15}N$ values for $NO_x$ from different emission sources and significant variations within each source (Fig. 1). Soil $NO_x$ has the lowest $\delta^{15}N$ values ranging from -59.8 ‰ to -19.8 ‰ (Li & Wang, 2008; Felix & Elliott, 2014; Yu & Elliott, 2017; Miller et al., 2018). The $NO_x$ emission from waste has the second-lowest $\delta^{15}N$ values, ranging from -29 ‰ to -8.5 ‰ (Felix & Elliott, 2014). The $NO_x$ emissions from vehicles are isotopically heavier relative to soil and waste, showing $\delta^{15}N$ values ranging from -19.2 ‰ to 17 ‰ (Moore, 1977; Heaton, 1990; Ammann et al., 1999; Pearson et al., 2000; Savard et al., 2009; Redling et al., 2013; Fibiger, 2014; Felix & Elliott, 2014; Walters et al., 2015a; Walters et al., 2015b). The $NO_x$ emissions from natural gas power plants are also isotopically heavier than soil and waste, showing $\delta^{15}N$ values ranging from -19.7 ‰ to -13.9 ‰ (Walters et al., 2015b). The $\delta^{15}N$ values of $NO_x$ emissions from coal-fired power plants have the highest values, ranging from 2.1 ‰ to 25.6 ‰ (Heaton, 1987; Heaton, 1990; Snape, 2003; Felix et al., 2012; Felix et al., 2015; Savard et al., 2017). The implement of emission control technology tends to increase $NO_x$ $\delta^{15}N$ values. The $\delta^{15}N$ value of $NO_x$ emitted from coal-fired power plant equipped with SCR ranges from 15.5 ‰ to 25.6 ‰ (Felix et al., 2012), the $\delta^{15}N$ of the $NO_x$ emissions from coal-fired power plant equipped with SNCR ranges from 13.6 ‰ to 15.1 ‰ (Felix et al., 2012), the $\delta^{15}N$ of the $NO_x$



emissions from coal-fired power plants equipped with OFA/LNB ranges from 9.0 ‰ to 12.6 ‰
(Felix et al., 2012). Similar isotope enrichment of $NO_x$ has been noted in vehicles as their catalytic
converters warm and become efficient (Walters et al, 2015a).
4        These distinctive differences in $\delta^{15}N$ values among different $NO_x$ emission sources suggest
$\delta^{15}N$ could be an effective tracer of atmospheric $NO_x$ sources. For example, Redling et al. (2003)
found higher $\delta^{15}N$ of $NO_2$ in samples collected closer to the highway compared to those adjacent
to a forest, showing the emissions from vehicles were dominant near the highway. In addition, a
strong positive correlation between the amount of $NO_x$ emission from coal-fired power plants
within 400 km radial area of study sites and $\delta^{15}N(NO_3^-)$ of wet and dry deposition has been
demonstrated (Elliott et al., 2007; 2009). What is lacking is a systematic way of connecting $\delta^{15}N$
values of $NO_x$ sources, regional emissions, and data from numerous studies to measurements of
$\delta^{15}N$ in $NO_y$.
13       Here we have simulated the emission of $^{15}NO_x$ and compared the predicted $\delta^{15}N(NO_x)$ values
with the recent measurements. The $\delta^{15}N$ values of atmospheric $NO_x$ are impacted by three main
factors. The first is the inherent variability of the $\delta^{15}N$ values of $NO_x$ emissions in time and space.
Secondly, atmospheric processes that mix the $NO_x$ emissions, blurring multiple emission sources
within a mixing lifetime relative to the $NO_x$ chemical lifetime (~1 day). And thirdly, isotope effects
occurring during tropospheric photochemistry may alter the $\delta^{15}N$ of $NO_x$ emissions as they are
transformed from $NO_x$ into $NO_y$. In this paper, we first consider the effects from the first
consideration, the variation in $NO_x$ emission sources over time and space. In two companion
papers, we will discuss the impacts from atmospheric mixing and tropospheric photochemistry by
using the emission simulation presented here as the input dataset for the Community Multiscale
Air Quality Modeling System (CMAQ) to simulate $\delta^{15}N$ of atmospheric $NO_x$. Thus, this research
examines the variability in $NO_x$ emissions over time and space in the Midwestern US and
calculates $^{15}N$ emissions in order to predict the spatial and temporal changes of $\delta^{15}N$ values of
emitted $NO_x$. The ultimate goal will be to evaluate the accuracy of the $NO_x$ emission inventory
using $^{15}N$.
## 2. Methodology
31       The EPA trace gas emission model SMOKE (Sparse Matrix Operator Kernel Emissions) was
used to simulate $^{14}NO_x$ and $^{15}NO_x$ emissions. $^{14}NO_x$ emissions we estimated using the SMOKE
model based on $NO_x$ emissions from 2002 NEI (National Emission Inventory, USEPA, 2014)
emission sectors and $^{15}N$ emission were determined using these emissions and the corresponding
$\delta^{15}N$ values of $NO_x$ sources from previous research (Table 1). Using the definition of $\delta^{15}N$ (‰),
$^{15}NO_x$ emitted by each SMOKE processing category (area, biogenic, mobile, and point) was
calculated by
$$^{15}NO_x(i) = {}^{14}NO_x(i) \times {}^{15}R_{NO_x}(i)$$

39                                                                               Eq. (2)
where $^{14}NO_x(i)$ are the $NO_x$ emissions for each category (i) obtained from NEI and SMOKE and
$^{15}R_{NOxi}$ is a $^{15}N$ emission factor ($^{15}NO_{Xi}/^{14}NO_{Xi}$) calculated by:
$$^{15}R_{NO_x}(i) = \left(\frac{\delta^{15}N_{NO_x}(i)}{1000} + 1\right) \times 0.0036$$

43                                                                               Eq. (3)
$\delta^{15}N_{NOx(i)}$ is the $\delta^{15}N$ value of some $NO_x$ source (i = area, biogenic, mobile, and point) and 0.0036
is the $^{15}N/^{14}N$ of air $N_2$, the reference point for $\delta^{15}N$ values. Thus, to use Eq. (2) we extended a



NO$_x$ emission dataset for the Midwestern US ($^{15}$NO$_x$ (i)) and used recent measurements to
determine δ$^{15}$N$_{NOx}$ values for major NO$_x$ emission sources ($^{15}$R$_{NOxi}$) by using Eq. (3).
Annual emissions estimates by 2002 NEI for the Midwestern United States was obtained from
NEI at the county-level and was converted into hourly emissions on a 12 km x 12 km grid over
the Midwestern United States and previously published (Spak, Holloway, & Stone, 2007). The
modeling domain includes latitudes between 37 º N and 45 º N, and longitudes between 98º W and
78º W, which fully covers the states of Minnesota, Iowa, Missouri, Wisconsin, Illinois, Michigan,
Indiana, Kentucky, Ohio, and West Virginia, and partially covers North Dakota, South Dakota,
Nebraska, Kansas, Tennessee, North Carolina, Virginia, Maryland, Pennsylvania, and New York.
On-road gasoline, on-road diesel, off-road gasoline, off-road diesel, coal-fired power plant, natural
gas power plant, soil, and livestock wastes are the main sources of NO$_x$ emissions in the NEI
(USEPA, 2014). These were imported into models that used parameters such as land use, plant
species, temperature, growing season, plume rise, roadway type, vehicle classification, and travel
time for vehicle emissions to convert them into hourly NO$_x$ emissions. SMOKE categorizes NO$_x$
emissions into four "processing categories": Biogenic, Mobile, Point, and Area (Table 1).
The choice of the 2002 version of NEI is, in part, arbitrary for several reasons. First, in order
to compare the model estimated δ$^{15}$N values with observations, it requires the emission inventory
to be relevant to the same timeframe as the δ$^{15}$N measurements of the NO$_y$. The data sets we
compare to the model (discussed below) span the late 1990's to 2009, thus the 2002 inventory is
more relevant than later inventories (2008 onward). Secondly, the current model is predicting the
initial δ$^{15}$N value, but this value will be altered by two effects. First, the role of atmospheric
transport and deposition, which will blur the regional δ$^{15}$N value of emissions based on emission
strength, mixing vigor, and deposition schemes. Secondly, photochemical and equilibrium isotope
effects that occur during the transformation of NO$_x$ into NO$_3^-$, which is the most of the available
NO$_y$ δ$^{15}$N data, measured from either rain or aerosols. Thus, it was not expected that this current
"emission only" model would accurately predict the δ$^{15}$N values of NO$_3^-$. Instead, the current work
is a proof of concept paper that addresses some basic questions, for instance, do we expect regional
and seasonal differences in δ$^{15}$N values of NO$_x$, and are they at least comparable to observations
in NO$_y$? We emphasize that the effects of atmospheric mixing and tropospheric photochemistry
will be addressed in subsequent papers.

| SMOKE Processing Category | NEI Sector | δ$^{15}$N-NO$_x$ (‰) from previous research | δ$^{15}$N-NO$_x$ (‰) choose for this study |
|---|---|---|---|
| Biogenic | Soil | -59.8 ~ -14.0 | -34.3 (Felix & Elliott, 2014) |
| Area | Livestock Waste | -29 ~ -8.5 | -18.8 (Felix & Elliott, 2014) |





| | | | |
|---|---|---|---|
| | Off-road Gasoline | -21.1 ~ 8.5 | -11.5 (Walters et al., 2015b) |
| | Off-road Diesel | | -10.5 (Walters et al., 2015b) |
| Mobile | On-road Gasoline | -28.1 ~ 17 | -2.7 (Walters et al., 2015b) |
| | On-road Diesel | | -2.5 (Walters et al., 2015b) |
| Point | Coal-fired Fossil Fuel Combustion | -19.7 ~ 25.6 | 15 (Felix et al., 2012) |
| | Natural Gas Fossil Fuel Combustion | | -16.5 (Walters et al., 2015) |

Table 1: The $\delta^{15}N$ values (in ‰) for $NO_x$ emission sources based on SMOKE processing category and NEI sector

## 2.1 Biogenic source of $NO_x$ emission

Biogenic sources of $NO_x$ are predominately by-products of microbial nitrification and denitrification occurring in soil. The Biogenic Emissions Inventory System (BEIS) was implemented within SMOKE to estimate hourly emissions from biogenic sources. The normalized emission was first generated based on 230 land-use types from the Biogenic Emission Landcover Database (USEPA, 2018), a normalized emission factor of $NO_x$, and land cover, to indicate the emission under standard environmental conditions (at 30 °C and 1000 µmol $m^{-2}$ $s^{-1}$ photosynthetic active radiation). Then, meteorological data generated by MM5 (Fifth-Generation Penn State/NCAR Mesoscale Model) (Grell, Dudhia, & Stauffer, 1994) was incorporated into BEIS and was used to finalize the speciated and temporally allocated emissions from biogenic sources by the algorithm for $NO_x$. This algorithm uses three steps. First, the land surface was designated by the land use as agriculture and non-agriculture based on Biogenic Emission Landcover Database. Second, $NO_x$ emissions were normalized based on temperature, precipitation, fertilizer application, and crop canopy coverage during the crop growing season (April 1 to October 31). Finally, for $NO_x$ emissions over agriculture areas during the non-growing season and $NO_x$ emissions over non-agriculture areas throughout the year, the emission $NO_x$ factor was limited to that for grassland, and the only temperature was used to normalize $NO_x$ emission (Pierce, 2001; Vukovich & Pierce, 2002; Schwede et al., 2005; Pouliot & Pierce, 2009; USEPA, 2018).

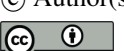



The $NO_x$ emission from the soil is regarded as a biogenic source in SMOKE, and there are
only a few measurements of the $\delta^{15}N$ values of biogenic $NO_x$. Li & Wang (2008) measured the
$NO_x$ fluxes using dynamic flow chambers for 2 to 13 days after cropland soil was fertilized by
either urea (n=9) or ammonium bicarbonate (n=9), and the $\delta^{15}N$ values of $NO_x$ ranged from -48.9
‰ to -19.8 ‰. Felix & Elliott (2014) placed a passive $NO_2$ sampler in a static flux chamber
installed in a cornfield. $NO_2$ was continuously collected from Jun 19-22, 2010 after 135 kg N/ha
of fertilizer was applied, and from Jun 2-19, 2011 after 40 kg N/ha of fertilizer application. The
$\delta^{15}N$ values of $NO_x$ emissions from these measurements range from -30.8 ‰ to -26.5 ‰. Miller et
al. (2018) used a static flux chamber to collect soil $NO_x$ emission 2~3 samples daily from May 17
to 26, 2016, and 2~4 samples daily from May 22 to Jun 3, 2017. The $\delta^{15}N$ values of $NO_x$ emissions
from these 37 samples ranged from -44.2 ‰ to -14.0 ‰. Yu & Elliott (2017) collected 15 samples
from soil plots for the $\delta^{15}N$ value of NO flux over a fallow field 2 weeks after the precipitation.
The $\delta^{15}N$ values of $NO_x$ emissions from these measurements range from -59.8 ‰ to -23.4 ‰, with
a standard deviation of ±11.25 ‰. The $\delta^{15}N$ values of $NO_x$ emissions from soil wetted with $NO_3^-$
aqueous solution treatments averaged -40.3 ± 0.75 ‰, while the $\delta^{15}N$ values of $NO_x$ emissions
from soil wetted with $NO_2^-$ aqueous solution treatments averaged -29.1 ± 4.17 ‰ suggesting there
are unique isotope effects for each step during $NO_3^- \rightarrow NO_2^- \rightarrow NO$ steps. The $\delta^{15}N$ values of $NO_x$
emissions from soil wetted with $NH_4^+$ aqueous solution treatments averaged -57.8 ± 1.91 ‰,
indicating $\delta^{15}N$ of $NO_x$ derived from nitrification is different than that from denitrification. Based
on these studies we adopted a $\delta^{15}N$ value for $NO_x$ emissions from the soil of -34.3 ‰, which is the
average value of these previous studies, to determine the emission rate of $^{15}NO_x$ from biogenic
sources using Eq. (2) and (3).
## 2.2 Mobile source of $NO_x$ emission
The emission of $NO_x$ based on on-road vehicle activity was estimated using MOBILE6, a
model developed by the EPA's Office of Transportation and Air Quality. There are three main
factors that are considered to estimate on-road vehicle $NO_x$ emissions. The first is the emission
rate per mile traveled for 28 different classifications of vehicles. The second is the emission factor
based on 10 different types of operating conditions (running, start, hot soak, diurnal, resting, run
loss, crankcase, refueling, brake wear, and tire wear), travel speed over 33 different road types
with distinct average speed, types of fuel being consumed, and ambient temperature. Finally, the
number of vehicles in each classification, emission type, and fuel type along with each type of
roadway during certain periods (USEPA, 2003; Houyoux, 2005). MOBILE6 and SMOKE were
used to determine $NO_x$ emissions along the roadways and were converted into hourly emissions
within each 12 km × 12 km grid cell.
The $NO_x$ emission from on-road vehicle employ an estimated $\delta^{15}N$ value from -28.1 ‰ to
+17 ‰ (Moore, 1977; Heaton, 1990; Ammann et al., 1999; Pearson et al., 2000; Savard et al.,
2009; Redling et al., 2013; Felix & Elliott, 2014; Fibiger, 2014; Walters et al., 2015a, 2015b). We
have excluded studies that infer $NO_x$ $\delta^{15}N$ by measuring plant proxies or passive sampling in the
environment (Ammann et al., 1999; Pearson et al.,2000; Savard et al. 2009; Redling et al., 2013;
Felix & Elliott, 2014). This is because of equilibrium and kinetic isotope effects that can occur as
$NO_x$ reacts in the atmosphere to form $NO_y$, prior to $NO_x$ deposition. In addition, the role vegetation
plays in $NO_x$ removal and atmospheric processes that mix the $\delta^{15}N$ of emission with the
surroundings can also alter the $\delta^{15}N$ from the mobile source. Instead, we estimated the $\delta^{15}N$ value
of $NO_x$ emissions from vehicles only using studies that directly measured tailpipe $NO_x$ emissions.



There is a handful of $NO_x$ $\delta^{15}N$ values measured from tailpipes, that span several decades.
Moore (1977) collected 3 samples of tailpipe $NO_x$ from one vehicle at different loads and engine
speeds, which had $\delta^{15}N$ values of $3.7 \pm 0.3$ ‰. Heaton (1990) collected 8 samples from the tailpipes
of 6 vehicles, on a testbed and on-road with different load and engine speeds. The resulting $\delta^{15}N$
values spanned -13 ‰ to 2 ‰, with an average of $-7.5 \pm 4.7$ ‰. Neither Heaton nor Moore noted
whether these 6 vehicles were equipped with any catalytic $NO_x$ reduction technology, but it is
unlikely since late 1970 and 80's s vehicles were seldomly equipped with catalytic $NO_x$ reduction
technology. Fibiger (2014) measured 5 samples of $NO_x$ from diesel engines without SCR emitted
into a smog chamber, the $\delta^{15}N$ values range from -19.2 ‰ to -16.7 ‰ ($\pm0.97$ ‰). The most
comprehensive studies on vehicle $NO_x$ $\delta^{15}N$ values are by Walters et al. (2015a, 2015 b). These
studies were chosen to assign the $\delta^{15}N$ of $NO_x$ emissions from vehicles in this study because these
measurements were taken directly from vehicle tailpipes, rather than inferring them (i.e from
roadside plant material, tree rings, or roadside $NO_2$) and had more samples (n = 73) compared to
other studies. In addition, it measured gas and diesel vehicles separately, including those with and
without three-way catalytic converter (TCC) and SCR technology. They also measured on-road
and off-road vehicles separately. This research showed that the $\delta^{15}N$ of $NO_x$ for vehicles without
SCR or when SCR was not functioning was negative, at around -15‰. As SCRs warmed and
became efficient at reducing $NO_x$ the $\delta^{15}N$ value became less negative and even went positive. The
measurements showed that the $\delta^{15}N$ values of $NO_x$ emitted by gasoline on-road vehicle averages
at $-2.5 \pm 1.5$ ‰, and on-road diesel ranged from -5 ‰ to 0 ‰.
The emission rate of $^{15}NO_x$ from the mobile source was determined by Eq. 4 grid by grid,
according to the contributions from on-road gasoline vehicles and on-road diesel vehicles, as well
as their corresponding $\delta^{15}N$ values of these two types of vehicles grid by grid. $NO_x$ emissions from
off-road vehicles are regarded as area sources in SMOKE, which were processed over each county.
In contrast, $NO_x$ emissions from on-road vehicles are regarded as the mobile source in SMOKE,
which will be processed along each highway. Each grid emission rate of $^{15}NO_x$ was assigned based
on the contributions from gasoline and diesel vehicles, as well as the relative $\delta^{15}N$ values. The
$\delta^{15}N$ of on-road gasoline vehicles ($-2.7 \pm 0.8$ ‰) was based on the average of the vehicle travel
time within each region with the same zip code (Walters et al., 2015b).
$$^{15}NO_x\,(mobile) = \left(\frac{\delta^{15}N_{NO_x\,(on-road\,gas)}}{1000} + 1\right) \times 0.0036 \times {^{14}NO_x}\,(on-road\,gas)$$
$$+ \left(\frac{\delta^{15}N_{NO_x\,(on-road\,diesel)}}{1000} + 1\right) \times 0.0036 \times {^{14}NO_x}\,(on-road\,diesel) \qquad \text{Eq. (4)}$$
Where $\delta^{15}N_{NO_x\,(on-road\,gas)} = -12.35 + 3.02 \times \ln(t + 0.455)$
## 2.3 Point source of $NO_x$ emission
The main NEI sectors for large amount of anthropogenic $NO_x$ emissions that are located at a
fixed, stationary position are categorized as $NO_x$ point sources. These include $NO_x$ emitted by
fugitive dust and power plants. Fugitive dust does not significantly contribute to point $NO_x$
emissions, so our inventory focused on power plants (Houyoux, 2005). Power plants were
separated into two different types: EGU (electric generating units) and Non-EGU (e.g. commercial
and industrial combustions). The emissions from EGUs account for 50-55% of the point source
$NO_x$ emissions, while non-EGUs account for 45-50%.
The $\delta^{15}N$ value of $NO_x$ emitted from power plants have been estimated to vary from -19.7 ‰
to 25.6 ‰ (Heaton, 1987; Heaton, 1990; Snape, 2003; Felix et al., 2012; Felix et al., 2015; Walters
et al., 2015b; Savard et al., 2017). We have ignored studies that measured $\delta^{15}N$ of $NO_3^-$ or $HNO_3$



from EGUs (Felix et al., 2015, Savard et al., 2017) and instead, only consider those studies that directly measured $\delta^{15}N$ of $NO_x$. Heaton (1990) collected 5 samples from the different coal-fired power stations with wall-fired and tangentially-fired boilers, at different power of 48, 500, and 600 MW. The $\delta^{15}N$ values of $NO_x$ emissions from these measurements range from 6 ‰ to 13 ‰, with a standard deviation of 2.9 ‰. Snape (2003) measured 36 samples from power plants using three different types of coals in combustion chars in a drop tube reactor. The $\delta^{15}N$ values of $NO_x$ ranged from 2.1 ‰ to 7.2 ‰, with a standard deviation of 1.37 ‰. The most comprehensive study on coal-fired power plants' $NO_x$ values was by Felix et al. (2012). They measured the $\delta^{15}N$ values of $NO_x$ emission from the coal-fired power stations with and without different emission control technologies. 16 coal-fired power plants with SCR, 3 coal-fired power plants with SNCR, 15 coal-fired power plants with OFA/LNB, and 8 coal-fired power plants without emission control technology were measured. The $\delta^{15}N$ values of $NO_x$ emissions from these 42 measurements range from 9 ‰ to 25.6 ‰, with a standard deviation of 4.51 ‰. The $NO_x$ $\delta^{15}N$ values when different emission control technologies were used varied: the $\delta^{15}N$ values of $NO_x$ emissions from coal-fired power plants with SCR range from 15.5 ‰ to 25.6 ‰, those with SNCR ranged from 13.6 ‰ to 15.1 ‰, and those with OFA/LNB ranged from 9.0 ‰ to 12.6 ‰. The $\delta^{15}N$ values of $NO_x$ emissions from coal-fired power plants without emission control technology range from 9.6 ‰ to 11.7 ‰, with a standard deviation of 0.79 ‰. According to Xing et al. (2013), about half of the coal-fired power plants in the United States are equipped with SCR. Thus, we assume 15 ‰ for the $NO_x$ emissions from coal-fired power plants, which is the average between SCR and other emission control technologies.

The most comprehensive study on natural gas-fired $NO_x$ values (Walters et al. 2015) collected 12 flue samples on the rooftop of a house from the ventilation pipe of a natural gas low-$NO_x$ burner residential furnace without $NO_x$ emission control technology. They also collected 11 flue samples from a sampling-port directly above a natural gas low-$NO_x$ burner power plant. The measurement showed that the $\delta^{15}N$ values of $NO_x$ emitted by natural gas power plants average $-16.5 \pm 1.7$ ‰, which we used for the $NO_x$ emission from natural gas power plants. The reason for using these values because they were measurements taken directly from the exhaust pipes, rather than inferring from downwind area or from rain samples, emitted by natural gas power plants, and included power plants with and without SCR technology. The latitude, longitude, and point sources characteristics (EGU and non-EGU, coal-fired or natural gas-fired, implementation of emission control technology) of each power plant was obtained from the US Energy Information Administration (2017). The power plants were assigned grids by their latitudes and longitudes, and the $\delta^{15}N$ values were assigned to these grids based on their emission characteristics, before determining the emission rate of $^{15}NO_x$ from point source using Eq. (2) and (3).

## 2.4 Area source of $NO_x$ emission

Area sources are the stationary anthropogenic $NO_x$ emissions that spread over a spatial extent and individually too small in magnitude to report as point sources. These include $NO_x$ emitted by off-road vehicles, residential combustion (anthracite coal, bituminous coal, distillate oil, residual oil, natural gas, liquified petroleum gas, and wood), industrial processes (chemical manufacturing, food, and kindred products, metal production, mineral processes, petroleum refining, wood products, construction, machinery, mining, and quarrying, etc), agriculture production (crops, fertilizer application, livestock, animal waste, etc), solvent utilization, storage and transport, waste disposal, treatment, and recovery, forest wildfires, as well as road dust and fugitive dust. Among





these, livestock and off-road vehicles are dominant, accounting for nearly 90% of area $NO_x$
emissions across the contiguous United States (Houyoux, 2005). The annual area emissions from
the NEI sectors were estimated at the county level and evenly divided into hourly emissions over
the 12 km × 12 km grid for use in chemical transport modeling.
The area $NO_x$ $\delta^{15}N$ values were based on the assumption that livestock waste and off-road
vehicles (utility vehicles for agricultural and residential purposes) accounted for total area sources.
Livestock waste $NO_x$ $\delta^{15}N$ values were taken from Felix & Elliott (2014) since it is currently the
only study about the $\delta^{15}N$ value of $NO_x$ livestock waste emissions. They placed passive sampler
with ventilation fans in an open-air and closed room in barns of cows and turkeys, respectively.
The $\delta^{15}N$ values of $NO_x$ emissions from these measurements range from -29 ‰ to -8.5 ‰. Among
these samples, the $\delta^{15}N$ of $NO_x$ emissions from turkey waste averages at -8.5 ‰, the $\delta^{15}N$ of $NO_x$
emissions from cow waste averages at -24.7 ‰. We used -18.8 ‰ as the values of $\delta^{15}N$ values for
$NO_x$ emissions from livestock waste, which is the weighted average of the $\delta^{15}N$ of $NO_x$ from turkey
waste and cow waste emissions, roughly based on the population of turkey and cows on farms
across the United States. We used the $\delta^{15}N$ values from Walters et al. (2015b) to estimate the $\delta^{15}N$
value of $NO_x$ emissions from the off-road vehicle since it is the latest in detail study that measured
the $\delta^{15}N$ value of $NO_x$ specifically from the off-road vehicle. They collected 45 samples from the
tailpipe of 9 different off-road vehicles (gasoline and diesel) with and without SCR, and before
and after the sufficient engine warm-up times. The measurement showed that the $\delta^{15}N$ values of
$NO_x$ emitted by gasoline-powered off-road vehicle averaged -11.5 ± 2.7 ‰, diesel off-road
vehicles without SCR averaged -19 ‰ ± 2 ‰, and diesel off-road vehicle with SCR averaged -2
‰ ± 8 ‰. The emission rate of $^{15}NO_x$ from area source was determined by Eq. 5 grid by grid,
according to the contributions from waste, off-road gasoline vehicle, and off-road diesel vehicle,
as well as their corresponding $\delta^{15}N$ values based on previous researches.

$$^{15}NO_x\ (area) = \left(\frac{\delta^{15}N_{NO_x\ (waste)}}{1000} + 1\right) \times 0.0036 \times {}^{14}NO_x\ (waste)$$

$$+ \left(\frac{\delta^{15}N_{NO_x\ (off-road\ gas)}}{1000} + 1\right) \times 0.0036 \times {}^{14}NO_x\ (off-road\ gas)$$

$$+ \left(\frac{\delta^{15}N_{NO_x\ (off-road\ diesel)}}{1000} + 1\right) \times 0.0036 \times {}^{14}NO_x\ (off-road\ diesel) \qquad \text{Eq. (5)}$$

The county-level annual $^{14}NO_x$ emission for the Midwestern US from NEI was converted to
the dataset with hourly $^{14}NO_x$ emission over 12 × 12 km grids throughout the year. During this
process, different NEI emission sectors were treated differently. Livestock waste and off-road
vehicles were regarded as area sources by SMOKE, of which the $^{14}NO_x$ emission over each county
was evenly divided into the grids. Power plants were regarded as the point source by SMOKE, of
which the $^{14}NO_x$ emission from these facilities was located into the corresponding grids according
to their latitudes and longitudes. On-road vehicles were regarded as the mobile source by SMOKE,
of which the $^{14}NO_x$ emission along the roadways was estimated by MOBILE model, based on
vehicle classifications, emission types, road type, fuel type, ambient temperature, and the number
of vehicles along each roadway during each hour, before evenly dividing $NO_x$ emission along each
roadway into groups of 12 × 12 km grids. The soil was regarded as the biogenic source by SMOKE,
of which the $^{14}NO_x$ emission produced by microbial nitrification and denitrification was estimated
by BEIS model, based on land use type, normalized emission factor of $NO_x$, land cover,
temperature, precipitation, fertilizer application, crop growing season, and crop canopy coverage
during the growing season, over each 12 × 12 km grid. Then, the $^{15}NO_x$ emission of each SMOKE





1     processing category was incorporated into the dataset based on the $\delta^{15}$N values from previous
2     research (Table 1) and Eq. (2-5).

3     $$\delta^{15}N_{NO_x\,(total)} = \left(\frac{\frac{^{15}NO_x\,(area)+^{15}NO_x\,(biog)+^{15}NO_x\,(mobile)+^{15}NO_x\,(point)}{^{14}NO_x\,(area)+^{14}NO_x\,(biog)+^{14}NO_x\,(mobile)+^{14}NO_x\,(point)}}{0.0036} - 1\right) \times 1000 \qquad \text{Eq. (6)}$$

## 3. Results and Discussion

### 3.1 Simulated spatial variability of NO$_x$ emission rates

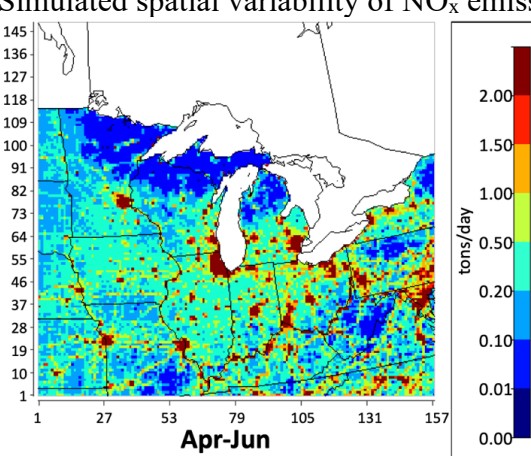

Figure 2: Total NO$_x$ emission in the Midwest between April and June in tons/day. High NO$_x$ emissions are associated with major urban areas such as Chicago, Detroit, Minneapolis-St Paul, Kansas City, St. Louis, Indianapolis, and Louisville.

We first examine the spatial heterogeneity of the NO$_x$ emission rate for a single time period to illustrate the overall pattern of NO$_x$ emission over the domain (Fig. 2). This is because the $\delta^{15}$N value of total NO$_x$ emission is determined by the fraction of each NO$_x$ source (Eq. 6), which in turn is a function of their emission rates. Since our NO$_x$ emissions are gridded by SMOKE using the NEI, they are by definition correct with respect to the NEI. However, a brief discussion of the salient geographic distribution of NO$_x$ emissions and comparisons with other studies is warranted for completeness and as a backdrop for the discussion of NO$_x$ fractions and resulting $\delta^{15}$N values. We have arbitrarily chosen to sum the NO$_x$ emissions during the April to June time period for this discussion.

The seasonal average NO$_x$ emissions within the geographic domain during April to June range from less than 0.01 tons N/day to more than 15 tons N/day, with the seasonal grid average of 0.904 tons/day. The average NO$_x$ emission over the 12 × 12 km grids simulated by SMOKE agrees well with estimates in previous studies, which was between 0.81 and 1.02 tons/day over the grids with the same size as this research but for the United States nationwide (Dignon & Hameed, 1989; Farrell et al., 1999; Selden et al., 1999; Xing et al, 2012). Within 75% of the grids within the geographic domain, the NO$_x$ emissions are relatively low, ranging from between 0 and 0.5 tons/day (Fig. S1). Geographically, these grids are located in rural areas some distance away from metropolitan areas and highways (Fig. 2). The NO$_x$ emission within about 20% of the grids is relatively moderate, ranging between 0.5 and 2.0 tons/day (Fig. S1). Geographically, these grids are mainly located along major highways and areas with medium population densities (Fig. 2). Urban centers comprise about 5% of the grids within the geographic domain and these have high



NO$_x$ emissions rates, ranging between 2.0 and 15.0 tons/day (Fig. S1). The metropolitan area's average is 5.03 tons/day, which is nearly 14 times of the average emission rate over the rest of the grids within the geographic domain (0.37 tons/day) due to high vehicle density associated with high population densities. The highest emissions rates are located within large cities (Fig. 2), such as Chicago, Detroit, Minneapolis-St Paul, Kansas City, St. Louis, Indianapolis, and Louisville, as well as the edge of the east coast metropolitan area (dark red). Summing the NO$_x$ emissions among the grids that encompass these major midwestern cities, yields city-level NO$_x$ emission rates that vary from 61.2 tons/day (Louisville, KY) to 634.1 tons/day (Chicago, IL). These city-level NO$_x$ emission rate simulated by SMOKE (Table 2) agrees well with estimates derived from the Ozone Monitoring Instrument (OMI) in a previous study (Lu et al., 2015). Grids containing power plants are the significant NO$_x$ hotspots within the geographic domain. These account for less than 1% of the grids within the geographic domain, but the NO$_x$ emissions from a single grid that contains a power plant could be as high as 93.4 tons/day. Geographically, the power plants are mainly located along the Ohio River valley, near other water bodies, and often close to metropolitan areas (Fig. 2). The NO$_x$ emission rates of the major power plants within the Midwest simulated by SMOKE (Table 3) match well with the measurement from Continuous Emission Monitoring System (CEMS) (de Foy et al., 2015; Duncan et al., 2013; Kim et al., 2009).

The geographic distribution of grid-level annual NO$_x$ emission density in our simulation agrees with the county-level annual NO$_x$ emission density discussed in the 2002 NEI booklet (Fig. S2; USEPA, 2018). For both grid-level emission density simulated by SMOKE and county-level emission density estimated by NEI, the relatively low values (less than 2.5 tons/mile$^2$) occur in the rural areas, especially located in the states of Minnesota, Iowa, Missouri, as well as the Plains states on the western edge of the domain. Similarly, the relatively moderate values (between 2.5 tons/mile$^2$ and 7.5 tons/mile$^2$) occur in the grids or counties that contain major highways; and the relatively high values (greater than 12.5 tons/mile$^2$) occurs in the grids or counties within metropolitan areas or in the grids or counties that contain power plants. Comparing the maps in different schemes, in addition, to show the geographic distribution of NO$_x$ emission density at different levels, the map of grid-level NO$_x$ emission density clearly shows locations of the objects with relatively high resolution, such as highways and power plant, as well as the more precise geographical range of metropolitan areas. The map of grid-level total NO$_x$ emission provides a clear view of spatial variation, and show the geographic location of major cities, highways, and power plants, while it has obvious limitations. First, some power plants share the same grids with metropolitan areas or highways, which also has relatively high NO$_x$ emission. As a result, it is hard to determine the dominant source for these grids. Similarly, among the grid with relatively low NO$_x$ emission, the map of total NO$_x$ emission cannot reveal the dominant source over these areas. In order to explore the composition of NO$_x$ emission, the $\delta^{15}$N value of total NO$_x$ emission is necessary.

| Urban Area | SMOKE-simulated emission rate | | OMI-derived emission rate |
|---|---|---|---|
| | tons/day | tons/hr | tons/hr |
| Chicago, IL | 634.074 | 24.42 | 23.3±9.7 |
| Detroit, MI | 288.617 | 12.026 | 18.7±7.8 |
| Indianapolis, IN | 72.487 | 3.021 | 3.1±1.3 |





| | | | |
|---|---|---|---|
| Kansas City, MO | 150.733 | 6.281 | 5.1±2.1 |
| Louisville, KY | 61.178 | 2.549 | 2.5±1.0 |
| Minneapolis, MN | 220.957 | 9.207 | 9.3±3.9 |
| St. Louis, MO | 99.953 | 4.165 | 4.9±2.0 |

2     Table 2: The seasonal average $NO_x$ emission rate for major cities in the Midwest

| Power Plant Site | SMOKE-simulated emission rate tons/day | CEMS-measured emission rate | |
|---|---|---|---|
| | | kt/yr | tons/day |
| Paradise, KY | 93.414 | 38.33 | 105.014 |
| New Madrid, MO | 65.777 | 23.09 | 63.260 |
| T. Hill Energy Center, MO | 38.686 | 11.95 | 32.740 |
| Kincaid, IL | 38.934 | 11.92 | 32.644 |
| Powerton, IL | 62.394 | 21.56 | 59.068 |
| Jeffrey Energy Center, KS | 59.339 | 21.39 | 58.603 |

5     Table 3: The seasonal average $NO_x$ emission rate for major power plants in the Midwest





Figure 3: The geographical distribution of the fraction of $NO_x$ emission from each SMOKE processing category (area, biogenic, mobile, point) over each grid throughout the Midwest between April and June based on NEI-2002.

3

We next examine the spatial heterogeneity of the $NO_x$ fraction from each source category (Fig. 3) for the same time period (April to June). Since the $\delta^{15}N$ value of total $NO_x$ is determined by the fractions of each $NO_x$ emission source over each grid (Eq. 6), it is important to understand where in the domain these fractions differ and why. The area sources, which mainly consist of off-road vehicles, agriculture production, residential combustion, as well as the industrial processes, which are individually too low in magnitude to report as point sources, are fairly uniform in their distribution across the domain. The SMOKE simulation shows that $NO_x$ emissions from area sources contribute an average $NO_x$ emission fraction ($f_{area}$) of 0.271 for total $NO_x$ emission and 0.290 for anthropogenic $NO_x$ emission within the Midwest from April to June. This is slightly higher than the fraction of 0.279 for annual anthropogenic $NO_x$ emissions over the Continental United States, estimated by 2002 NEI (USEPA, 2018). The fractions of $NO_x$ emission from area sources over each grid cell within the geographic domain show a clear spatial variation. The area sources account for $NO_x$ emission fractions ranging from 0.125 to 0.5 over about 75% of the grids within the geographic domain (Fig. S3). Geographically, the grids with relatively higher $f_{area}$ are located in the rural area away from highways, especially in the states of Indiana, Illinois, Iowa, Minnesota, and Ohio, where agricultural is the most common land use classification. In the states of Wisconsin and Missouri, the $f_{area}$ is slightly lower due to the higher fraction of $NO_x$ emission from biogenic sources ($f_{biog}$). In the states of Pennsylvania and Michigan, the $f_{area}$ is slightly lower due to the higher fraction of $NO_x$ emission from mobile sources ($f_{mobile}$). In addition, the grids with $f_{area}$ greater than 0.75 are mainly located along the Mississippi River and Ohio River, where the demand for water consumption and wastewater discharging from agriculture production could be satisfied.

The fraction of biogenic $NO_x$ ($f_{biog}$) that are predominately by-products of microbial nitrification and denitrification occurring in soil, shows the clear spatial variation and is highest (from April to June) in the western portion of the domain (Fig. 3). The SMOKE simulation estimates that the fraction of biogenic $NO_x$ emission averages 0.065 within the Midwest from April to June. The biogenic $NO_x$ fraction is less than 0.5 in more than 90% of the grids within the geographic domain (Fig. S3). Geographically, the grids with relatively high $f_{biog}$ are located in the western regions of the Midwest, away from cities and highways, in the states of Minnesota, Iowa, Missouri, Wisconsin, and Illinois, where the density of agricultural acreage and natural vegetation is higher than other states. Furthermore, within regions with higher $f_{biog}$, the obvious low $f_{biog}$ values occur in the megacities and along the highways, which agrees well with the land-use related to the biogenic emission.

The SMOKE simulation shows that the $NO_x$ emissions from mobile sources contribute to the fraction ($f_{mobile}$) of 0.325 for total $NO_x$ emission and 0.347 for anthropologic $NO_x$ emission within the Midwest from April to June, which is slightly lower than the fraction of 0.380 for annual anthropologic $NO_x$ emission over the Continental United States, estimated by 2002 NEI (USEPA, 2018). The fractions of $NO_x$ emission from the mobile source over each grid cell within the geographic domain show a clear spatial variation. The value of $f_{mobile}$ within the geographic domain distributes evenly on the histogram (Fig. S3). Geographically, the grids with relatively higher $f_{mobile}$ are located in major metropolitan regions and along the highways, where vehicles have the highest density, especially in the states of Pennsylvania, New York, Virginia, West Virginia, and North Carolina. In addition, within the states with lower $f_{mobile}$, the obvious high $f_{mobile}$ values occur in the megacities and along the highways, which agrees well with the vehicle activities (US Census Bureau, n.d.).



The point sources consist mainly of EGUs, as well as commercial and industrial processes
involving combustion. Based on the SMOKE simulation, the $NO_x$ emission from point sources
contributes to the fraction ($f_{point}$) of 0.339 for total $NO_x$ emission and 0.363 for anthropologic $NO_x$
emission within the Midwest from April to June, which is slightly higher than the fraction of 0.343
for annual anthropologic $NO_x$ emission over the Continental United States, estimated by 2002 NEI
(USEPA, 2018). The fractions of $NO_x$ emission from the point source over each grid cell within
the geographic domain show a clear spatial variation. Geographically, the $NO_x$ emission from point
sources is dominant at the grids, where the power plants are located, mainly along the Ohio River
valley and near other water bodies close to metropolitan areas. The point sources have no
contribution to the $NO_x$ emission among about 96% of the grids within the geographic domain.
The rest of the 4% of the grids within the geographic domain are the locations of power plants.
About 1/4 of the power plants are not at the same grids as highways, thus these grids have a fraction
of at least 0.9 $NO_x$ emission from point sources. Whereas the other 3/4 of the power plants share
the same grids with highways, thus the point sources become relatively less dominant, due to the
dilution by the $NO_x$ emission from mobile sources.

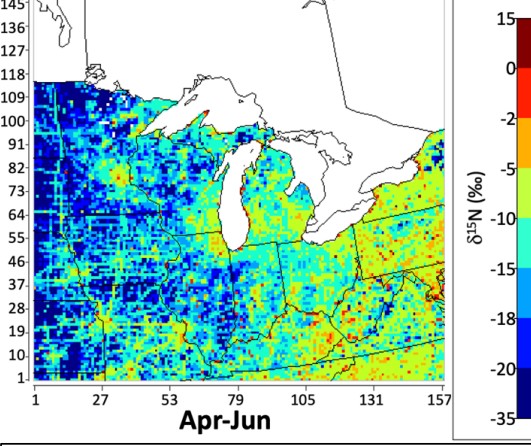

Figure 4: The $\delta^{15}N$ values of $NO_x$ emission during April-June are presented by color in each grid. The warmer the color, the higher $\delta^{15}N$ values of $NO_x$ emission.

Using these $NO_x$ emission source fractions in each grid, the $\delta^{15}N$ values of $NO_x$ were
simulated. We then examine the spatial heterogeneity of $\delta^{15}N$ values of $NO_x$ for a single time
period and interpret them in terms of changes if $NO_x$ emission fractions over the domain. The
predicted $\delta^{15}N$ values of $NO_x$ range from -35 ‰ to +15 ‰, with the seasonal average over the
Midwest of -13.18 ‰ during the April to June period. The $\delta^{15}N$ value of total $NO_x$ emissions in
the Midwest during the April to June period has a significant spatial variation (Fig. 4). This can be
qualitatively explained based on which emission source is dominant in a particular grid cell or
grouping of cells in a certain region. The $NO_x$ $\delta^{15}N$ model clearly shows the locations of big cities
such as Chicago, Detroit, Minneapolis-St Paul, Kansas City, St. Louis, Indianapolis, and Louisville
(gold and green). Likewise, major highways that connect these cities are obvious features (also
gold and green), particularly on the western side of the domain. This is a consequence of the fact
that in both cities and on major roads, on-road vehicles are the dominant $NO_x$ source with assigned





$\delta^{15}$N values of -2.5 ‰. In these grids, the NO$_x$ $\delta^{15}$N typically ranges from -5 to -10 ‰. Likewise, in the western part of the domain in the Midwest-Plains state region, where urban centers and population density is sparse and power plants are less numerous, soil emissions, with a $\delta^{15}$N value of -34.3‰, control the NO$_x$ budget. The predicted NO$_x$ $\delta^{15}$N values in these areas are very negative (dark blue), ranging from -20 to -34‰. In other grids, there are mixtures of sources such as mobile and biogenic leading to $\delta^{15}$N values in the negative teens (aqua color), which is a mixture between the agricultural and urban NO$_x$ sources. Similarly, the very positive $\delta^{15}$N grids (~ +15 ‰) are located in grids that contain major power plants that dominate the NO$_x$ emission budget (red and dark red), such as the Ohio River valley and West Virginia. These results show that there should be strong regional dependence on NO$_x$ $\delta^{15}$N values in the Midwestern United States.





1 ## 3.2 Seasonal variation in $\delta^{15}N$ of $NO_x$



Figure 5: The geographical distribution of the $\delta^{15}N$ value of total $NO_x$ emissions in each season (Winter: Jan-Mar; Spring: Apr-Jun; Summer: Jul-Sep; Fall: Oct-Dec) in per mil (‰) throughout the Midwest simulated by SMOKE, based on NEI-2002.

2        We next examine the temporal heterogeneity of $NO_x$ $\delta^{15}N$ values over the domain and
3 interpret them in terms of changes if $NO_x$ emission fractions as a function of time. The predicted
4 $\delta^{15}N$ value of total $NO_x$ emissions in the Midwest during each season shows a significant temporal



variation (Fig. 5). The $\delta^{15}$N values of NO$_x$ range from -35 ‰ to 15 ‰, with the annual average
over the Midwest at -6.15 ‰. The maps for different seasons show the obvious changes in $\delta^{15}$N
values over western regions of the Midwest, from green ($\delta^{15}$N = -15 ~ -5 ‰) to dark blue (-35 ~ -
15 ‰) during the month from April to October.
5       In order to qualitatively analyze the changes in $\delta^{15}$N(NO$_x$) among each season, the
distributions of $\delta^{15}$N(NO$_x$) among the same cut-offs as the maps on Fig. 5 were shown in the
histograms (Fig. S4). The grids with $\delta^{15}$N(NO$_x$) between -35‰ and -18‰ increase dramatically
from less than 10% during fall (Oct-Dec) and winter (Jan-Mar) to more than 20% during spring
(Apr-Jun) and summer (Jul-Sep). The grids with $\delta^{15}$N(NO$_x$) between -18‰ and -2‰ decrease from
around 90% during fall and winter to around 75% during spring and summer. In addition, the
distribution of $\delta^{15}$N(NO$_x$) shifts to lower values during spring and summer.
The significant temporal variation in the $\delta^{15}$N value of total NO$_x$ during different seasons can
be quantitatively explained by changing fractions of NO$_x$ emission from the biogenic source in
any grid (Fig. 6) using Eq. (6). Unlike other NO$_x$ emission source (figure not shown), the fraction
of NO$_x$ emission from biogenic sources changes significantly among each season within the
geographic domain, especially over the rural areas of the states of Minnesota, Iowa, Missouri,
Wisconsin, Illinois, Indiana, Kentucky, Michigan, and Ohio (Fig. 6). The fraction of NO$_x$ emission
from biogenic sources over these areas increases from less than 0.25 to more than 0.50 during the
month from April to October, which is the growing season of the plant. During this period, the
surface temperature and precipitation are relatively higher. As a result, the canopy coverage of the
plants becomes higher, which leads to the increase of the NO$_x$ emission from biogenic sources
(Pierce, 2001; Vukovich & Pierce, 2002; Schwede et al., 2005; Pouliot & Pierce, 2009; USEPA,
2018). Besides this, the fertilizer application during this period is also responsible for the increase
in soil NO$_x$ emission (Li & Wang, 2008; Felix & Elliott, 2014).
In order to qualitatively analyze the changes in the fraction of NO$_x$ emission from biogenic
sources among each season, the distributions of the fractions among the same cut-offs as the maps
on Fig. 6 were shown in the histograms (Fig. S5). Comparing the distributions of the fractions of
NO$_x$ emission from biogenic sources among the histograms for each season, the effects from the
increasing of biogenic NO$_x$ emission during the growing season of plants are clearly shown. In
general, the distribution of the fraction shifts to higher values during spring (Apr-Jun) and summer
(Jul-Sep), indicating the increase of biogenic emission. As a result, the distribution of $\delta^{15}$N(NO$_x$)
shifts to lower values during the same period (Fig. 5). The percentage of the grids with the fraction
of biogenic emission less than 0.125 decreases dramatically from more than 50% during fall (Oct-
Dec) and winter (Jan-Mar) to less than 35% during spring (Apr-Jun) and summer (Jul-Sep). As the
NO$_x$ emission from biogenic source becomes dominant, the percentage of the grids with $\delta^{15}$N(NO$_x$)
between -35‰ and -18‰ increases, while the percentage of the grids with $\delta^{15}$N(NO$_x$) between -
18‰ and -2‰ decreases, which sufficiently explains the trends shown on Fig. 5.



Figure 6: The geographical distribution of the fraction of NO$_x$ emission from biogenic sources over each grid in each season (Winter: Jan-Mar; Spring: Apr-Jun; Summer: Jul-Sep; Fall: Oct-Dec) throughout the Midwest simulated by SMOKE, based on NEI-2002.

## 3.3 Different versions of emission inventories

3    The NO$_x$ budget estimated by different versions (years) of the emission inventory varies. In
4    order to compare the spatial heterogeneity of the fraction of NO$_x$ from each source category for
5    different emission inventory versions, the same analysis was done on the 2016 version of NEI (Fig.



7). Overall, the anthropologic $NO_x$ emission in the 2016 NEI is lower than in 2002, whereas the $NO_x$ emission from biogenic emission is higher, especially in the western part of the domain. The difference in temperature, precipitation, fertilizer application, and crop canopy coverage during the crop growing season, as well as the adjustments of the algorithms for different versions of BEIS, potentially cause the variation in the fraction of $NO_x$ emission from biogenic sources. The fraction of $NO_x$ emission from area source in the 2016 NEI was lower than 2002 NEI for most of the grids within the domain, except the hotspots in West Virginia, northern Michigan, and eastern Kansas. The 2016 fraction of $NO_x$ emission from the mobile source was lower than the 2002 NEI for most of the grids, especially in the eastern part of the domain. The fraction of $NO_x$ emission from point source based on 2016 NEI shows fewer hotspots comparing 2002 NEI, which indicates less amount of power plant operated within the domain. The implementation of $NO_x$ emission control technologies (SCR, SCNR, LNB, OFA), as well as the adjustments of the algorithms for different versions of MOBILE and MOVES, potentially cause the variation in the fraction of $NO_x$ emission from anthropologic sources. Due to the significantly higher fraction of $NO_x$ emission from biogenic source (Fig. S6) comparing to the estimation from 2002 NEI, the $\delta^{15}N$ value of total $NO_x$ based on 2016 NEI was lower (Fig. S7).





Figure 7: The geographical distribution of the fraction of NO$_x$ emission from each SMOKE processing category (area, biogenic, mobile, point) over each grid throughout the Midwest between April and June, based on NEI-2016.

2
3
4



## 3.4 Model-observation comparison

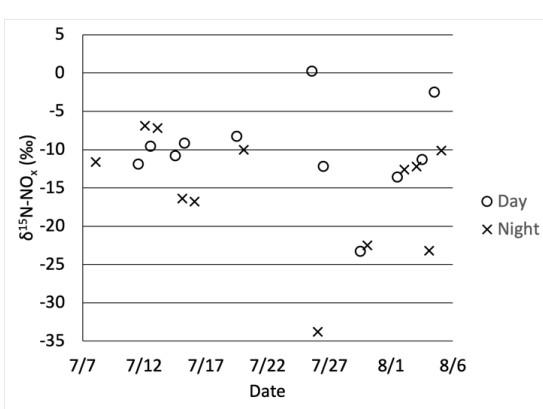

Figure 8: The $\delta^{15}N(NO_x)$ values measured at West Lafayette, IN between July 9 and August 5, 2016, from 8 am to 4 pm during the daytime (○), and from 9:30 pm to 5:30 am during the nighttime (×)

In order to evaluate the SMOKE simulation of Midwestern $\delta^{15}N(NO_x)$ values, they were
compared to several existing observational datasets. The first comparison is to the only direct
measurements of $\delta^{15}N(NO_x)$ within the domain, which occurred in West Lafayette, IN (Walters,
Fang, & Michalski, 2018). The West Lafayette, IN site is in the northwest part of Indiana and is
an NADP (National Atmospheric Deposition Program) site and home to Purdue University. 30
$NO_x$ samples were collected using denuder tubes between July 8 and August 5, 2016 (Fig. 8) from
8 am to 4 pm during the daytime, and from 9:30 pm to 5:30 am during the nighttime. The measured
$\delta^{15}N$ values of $NO_x$ in West Lafayette ranged from -23.3 to 0.2 ‰ during the daytime and ranged
from -33.8 to -6.9 ‰ during the nighttime.





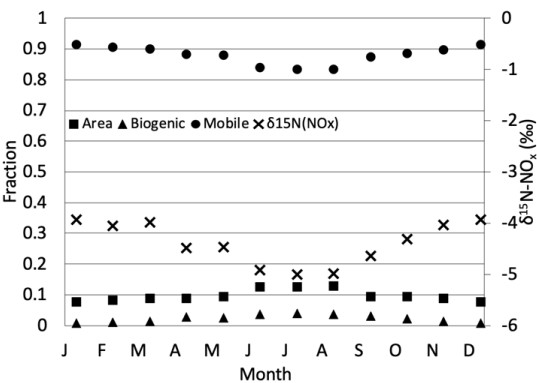

Figure 9: Fraction of monthly total $NO_x$ emission by each SMOKE processing category (area [■], biogenic [▲], mobile [●]), and the monthly $\delta^{15}N$ values of total $NO_x$ emission over the 12-km grid (right axis) over the 12-km grid that covers West Lafayette, IN simulated by SMOKE, based on NEI-2002.

The simulated $\delta^{15}N$ values of $NO_x$ in West Lafayette show trivial monthly variations, and a small 1‰ seasonal trend (Fig. 9, right axis). The simulation shows that the $\delta^{15}N$ values stay around -4 ‰ from January to March, start to decrease in April until reaching -5 ‰ in June, and then start to increase in September until returning to -4 ‰ in November. These $\delta^{15}N(NO_x)$ reflect that in West Lafayette mobile (on-road vehicle) is the dominant $NO_x$ source (Fig. 9, left axis). The $NO_x$ fraction from the mobile sector was between 0.8 and 0.9 throughout the year. Mobile $NO_x$ during summer is 10 % lower than average, which could be explained by the decrease in vehicle traffic during the summer holiday, when most students return to their home and when biogenic and area sources slightly increase due to peak agriculture activity. This seasonal change in fractions results in the -1‰ over the summer period.



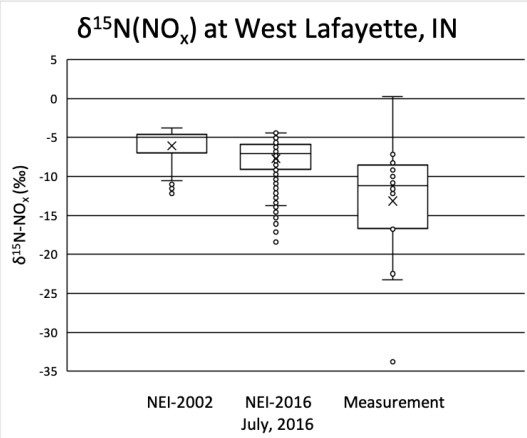

Figure 10: The distributions of $\delta^{15}N(NO_x)$ values over the 12-km grid that covers West Lafayette, IN from July 8 to August 5, simulated by SMOKE, using NEI-2002 (left) and NEI-2016 (middle) as the input, compare with the corresponding measurement (right) taken on July to August in 2016 (box: lower quartile, median, upper quartile; whisker: lower extreme, upper extreme; dots outside the whisker: outliers)

The SMOKE simulation of $\delta^{15}N$ values in West Lafayette, IN was compared with the measurement (Walters, Fang, & Michalski, 2018) from July 8 to August 5, 2016 (Fig. 10). The range of SMOKE simulated $\delta^{15}N(NO_x)$ from NEI-2002 ranges from -12.2‰ to -3.8‰, which is within the range of the corresponding measurement (-33.8 ~ 0.2 ‰). Whereas, the median (-5.0 ± 2.2 ‰) of SMOKE simulated $\delta^{15}N(NO_x)$ is higher than the median (-11.2 ± 8.0 ‰) of the measured values. As mentioned in section 3.3, the estimation of $NO_x$ emission from biogenic sources by NEI-2016 is higher than the estimation by NEI-2002. As a result, using the data in NEI-2016 as the input, SMOKE simulated $\delta^{15}N(NO_x)$ values are lower, with the median (-7.0 ± 2.4 ‰) and range (-18.4 ~ -4.4 ‰) closer to the corresponding measurement. By comparing the SMOKE simulated $\delta^{15}N(NO_x)$ with the corresponding measurements, the $NO_x$ emission budget in West Lafayette, IN, estimated by NEI-2016 is more accurate. While, the SMOKE simulated $\delta^{15}N(NO_x)$ values in West Lafayette, IN, based on both versions of NEI are higher than the corresponding measurements. Therefore, the emission from the soil, livestock waste, off-road vehicles, and natural gas power plant might be underestimated, and/or the emission from the on-road vehicle and coal-fired power plant might be overestimated for both versions of NEI.

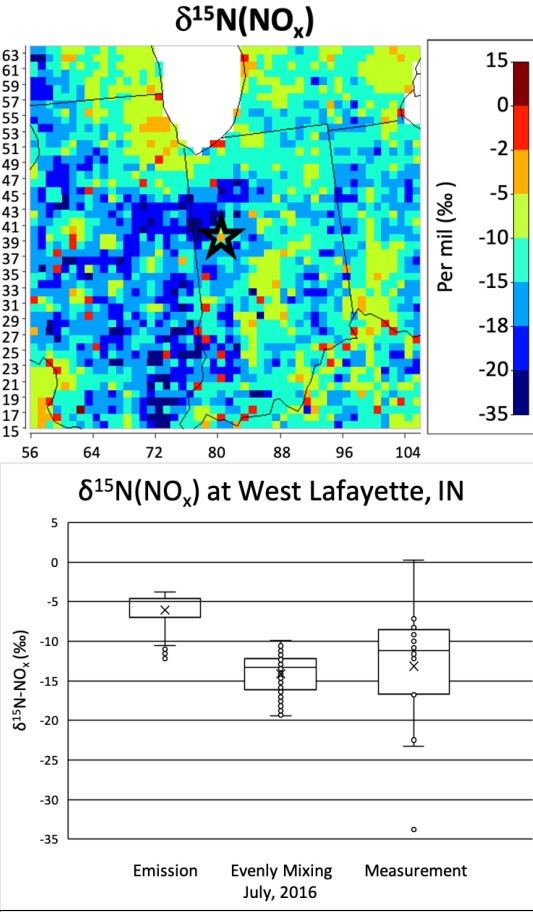

Figure 11: The $\delta^{15}N$ value of annual total $NO_x$ emissions in 12 $km^2$ grids (top), center on West Lafayette, IN (☆). The modeled (with and without mixing) and measured $\delta^{15}N(NO_x)$ distributions for West Lafayette between from July 8 to August 5 (bottom). (box: lower quartile, median, upper quartile; whisker: lower extreme, upper extreme; dots outside the whisker: outliers)

In addition to the effects from $NO_x$ emission sources, the lower values and greater variations in measured $\delta^{15}N(NO_x)$ might also be caused by the atmospheric mixing with the emission from surrounding grids, driven by the atmospheric processes. The map shows that the $NO_x$ emission around West Lafayette is isotopically lighter than the neighborhood emission (Fig. 11). Thus, the mass-weighted average of the emission within 24 grids around West Lafayette, IN was used to calculate the $\delta^{15}N(NO_x)$ values, which considered the equal mixing of the emissions from the neighborhood, driven by 4 m/s of wind speed (National Centers for Environmental Information, 2019) during the 0.84 days of atmospheric $NO_x$ lifetime (Stavrakou et al., 2013) (Eq. (7)). Using



this method, the simulated $\delta^{15}N$ values (median: -13.3 ± 2.5 ‰, range: -19.4 to -10.0 ‰) during
the study period was closer to the measured values (median: -9.7 ± 7.6 ‰, range: -31.4 ~ 0.4 ‰)
(Walters, Fang, & Michalski, 2018). Therefore, the $\delta^{15}N$ values are sensitive to effects from
neighborhood emissions (Fig. 11). The more appropriate method will be tested on CMAQ (The
Community Multiscale Air Quality Modeling System) in later researches, which takes the detailed
atmospheric conditions into account for the atmospheric mixing of the pollutants.
$$(\delta^{15}N_{NO_x})_{total} = \sum f_{grid\,(i)} \times \delta^{15}N_{grid(i)} \hspace{4cm} \text{Eq. (7)}$$



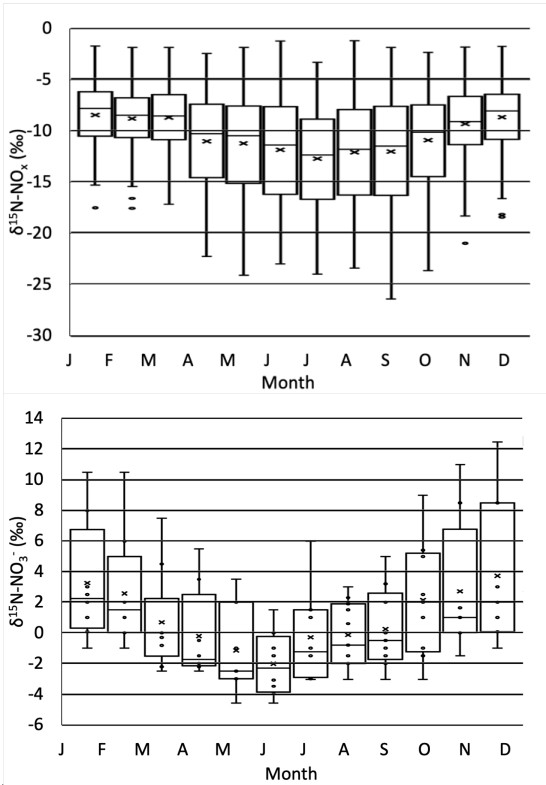

Figure 12: The SMOKE predicted $\delta^{15}N$ value of total $NO_x$ at 82 NADP sites (top) using NEI-2002 compared to the measured $\delta^{15}N$ of rain $NO_3^-$ (bottom) from prior studies.

Finally, we compared the emission model's predicted $NO_x$ $\delta^{15}N$ values at 82 NADP sites in
the Midwest (Fig. S8) with measurements of $NO_3^-$ $\delta^{15}N$ values (Elliott et al., 2009; Garten, 1992;
Hall et al., 2016; Occhipinti, 2008; Russell et al., 1998). The $\delta^{15}N$ values of $NO_x$ simulated by
SMOKE at these sites show large monthly variations and a seasonal trend (Fig. 12, top). The
monthly boxes are the 1$^{st}$ and 3$^{rd}$ quantiles of the simulated monthly $\delta^{15}N$ of $NO_x$ emissions at the
82 sites. The whiskers represent the minimum and maximum values without outliers. There is a
wide range of $\delta^{15}N(NO_x)$ values within each month, with a minimum during March (-17.1~ -1.9
‰) and a maximum during September (-26.5~-1.9 ‰). The seasonal trend shows low $\delta^{15}N(NO_x)$
during summer, with the median around -12 ‰, and high $\delta^{15}N(NO_x)$ during winter, with the
median around -8 ‰. The SPSS analysis result shows the monthly change of $\delta^{15}N$ values is
dominantly affected by biogenic emission. The effect from point sources is minimal since most of
the NADP sites are more than 12 km (grid size of SMOKE) away from the power plant. The NADP
sites are not in big cities but close to soil emission. Thus, biogenic emission has the strongest effect
on the $\delta^{15}N$ values of $NO_x$ emission, account for 86.6% of the change on $\delta^{15}N(NO_x)$.
16       Comparing with the SMOKE simulation, the measurements of $\delta^{15}N$ values of $NO_3^-$ in the
United States from previous researches (Elliott et al., 2009; Garten, 1992; Hall et al., 2016;
Occhipinti, 2008; Russell et al., 1998) shows the similar monthly variations and seasonal trend



(Fig. 12, bottom). There is a wide range of $\delta^{15}N(NO_3^-)$ values within each month, with a minimum during June (-4.6~ 1.5 ‰) and a maximum during December (-1.0~12.5 ‰). The seasonal trend shows low $\delta^{15}N(NO_3^-)$ during summer, with the median around -2 ‰, and high $\delta^{15}N(NO_3^-)$ during winter, with the median around 2 ‰. The measured $\delta^{15}N$ values of $NO_3^-$ has the same seasonal trend as the SMOKE simulated $\delta^{15}N$ values of $NO_x$. However, the measured $\delta^{15}N$ values of $NO_3^-$ is about 10 ‰ higher than the SMOKE simulated $\delta^{15}N$ values of $NO_x$. This is because of the photochemical and equilibrium isotope effects that occur during the transformation of $NO_x$ into $NO_3^-$, which enriches the $^{15}N$ isotopes in $NO_3^-$, as a more oxidized form of $NO_y$ (Walters & Michalski, 2015; Walters et al., 2016). The 10‰ difference between the measured $\delta^{15}N(NO_3^-)$ and the SMOKE simulated $\delta^{15}N (NO_x)$ agree well with the previous study (Chang et al., 2018). The effect of tropospheric photochemistry, including the net N isotope effect during the conversion of $NO_x$ to $NO_3^-$, will be addressed in subsequent papers.

## 4. Conclusion

The $\delta^{15}N$ of atmospheric $NO_x$ was simulated by SMOKE, by considering the $NO_x$ emissions from NEI emission sectors and the corresponding $\delta^{15}N$ values from previous researches. $\delta^{15}N$ is a decent tool to present the spatial and temporal composition of atmospheric $NO_x$, as well as the corresponding variation in $NO_x$ emission sources. The simulation indicates that the $NO_x$ emission from biogenic sources is the key driver for the variation of $\delta^{15}N$, especially among the NADP sites. Comparing with the measurements of $\delta^{15}N(NO_3^-)$ from NADP sites within the Midwest region, the simulated $\delta^{15}N$ agreed well with the seasonal trend and monthly variation. While, the simulated $NO_x$ is slightly heavier than the corresponding measurements in West Lafayette, IN, taken from July to August 2016. According to the previous researches, the uncertainty of $NO_x$ emission is 71-250% from soil and 10-15% from the vehicle. The variations among the removal efficiency of different emission control technologies vary from 30% to 90%, also causes the uncertainty of power plant $NO_x$ emission. In addition, in this study, due to the lack of measurements, the $\delta^{15}N$ of coal-fired and natural gas non-EGUs (industrial boilers, commercial and residential fuel combustions) were assumed to be the same as the $\delta^{15}N$ of coal-fired and natural gas EGUs respectively. Thus, detailed measurements of the $\delta^{15}N$ of non-EGUs are necessary for future study. Besides this, the non-road vehicles (aircraft, ships, and trains) also need to be included in the future study.

If we only consider the effects from $NO_x$ emission sources, the emission from soil, livestock waste, off-road vehicles, and natural gas power plant in West Lafayette, IN are possible to be underestimated, and the emission from the on-road vehicle and coal-fired power plant in West Lafayette, IN are possible to be overestimated. Another reason causing the estimated $NO_x$ isotopically heavier than measured $NO_x$ is the mixing caused by atmospheric processes, since the $NO_x$ emission from the surrounding region of West Lafayette, IN is lighter. In addition, the tropospheric photochemistry could also alter the $\delta^{15}N$ values during the processes that convert $NO_x$ to $NO_y$. The future work will explore the impacts of atmospheric processes and tropospheric photochemistry by incorporating $^{15}N$ into CMAQ and comparing the simulations with the corresponding measurements.

**Data availability:** The in-detail simulation results for $\delta^{15}N$ of $NO_x$ emission based on 2002 and 2016 versions of National Emission Inventory and the associated python codes are achieved on Zenodo.org (10.5281/zenodo.4048992).





**Author contributions:** Huan Fang and Greg Michalski were the investigator for the project and organized the tasks. Huan Fang develop the model codes and performed the simulation to incorporate $^{15}N$ into SMOKE outputs and generated $\delta^{15}N$ values. Greg Michalski helped Huan Fang in interpreting the results. Huan Fang prepared the manuscript with contributions from all co-authors.

**Acknowledgments:** We would like to thank the Purdue Research Foundation and the Purdue Climate Change Research Center for providing funding for the project. We would like to thank Scott Spak from School of Urban & Regional Planning, University of Iowa for simulating SMOKE using NEI-2002. We would like to thank the CMAS (Community Modeling and Analysis System) Data Warehouse for providing SMOKE input and output datasets based on NEI-2016.



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
