# Peer review of "Incorporating 15N into the outputs of SMOKE version 4.6 as the emission 1"

_Geoscientific Model Development, 2020_

## Short Comment (SC1) · 26 Feb 2021

Dear authors,

in my role as executive Editor I need to point out, that all GMD articles of type "development and technical paper" require the provision of the code used. Therefore it is

required, that you add a Code Availability Section to your article, stating how to access the exact SMOKE and CMAQ code versions used for this publications.

Best regards, Astrid Kerkweg

---

## Referee Comment (RC1) · Anonymous Referee #1 · 4 Mar 2021

In this paper by Fang and Michalski, the authors incorporated N isotope signatures of various NOx sources into the US EPA trace gas emission model SMOKE to simulate spatial and temporal variability of ambient d15N-NOx in the US Midwest region. Although comparisons between simulated and measured ambient d15NOx do not provide direct evidence for NOx source partitioning, due to the atmospheric mixing effect

and isotopic fractionations as pointed out by the authors, this work is an important initial step toward better use of NOx isotopes to resolve uncertainties in local and regional NOx emission inventories. I have two questions.

First, it seems that the authors did not consider the uncertainties associated with emission inventories and the d15N signatures in their simulation. For example, the NEI inventories are known to contain large errors, especially for biogenic sources. As pointed out by the authors, the d15N signatures are also highly uncertain and span large ranges for individual sources. However, only an average value was used for each source. What if the d15N source signatures varied over space and time? Would this variability in the source signatures change significantly the simulated spatial and temporal patterns?

The second question I had is regarding the plant canopy effect on biogenic NOx removal. As mentioned by the authors, soil-emitted NOx can be removed by overlying canopies to a large extent (up to 75%). However, this effect was not considered in the simulation. I am curious to see if the simulated d15N patterns would be changed by explicitly considering this canopy effect.

————————————————————

---

## Referee Comment (RC2) · Anonymous Referee #2 · 24 Mar 2021

This manuscript details the incorporation of N isotopic signature of NOx into SMOKE and the simulation results. It is nice to see the progress made towards using isotopic signatures to constrain the NOx inventory. Below are my comments:

Major remarks 1. The authors adopted a mean d15N value for each source as the
model input. However, large uncertainties are associated with each of the sources. 2. Page 8, lines 40-43, reasons were given here about why the results from passive sampling were excluded. What are the reasons why the passive sampling results were adopted for soil emissions. 3. Page 9, lines 20-31, I am confused about how the d15N-NOx of gasoline values were chosen, as the authors show two different ones -2.5±1.5‰ and -2.7±1.8‰ and also show an equation in line 32. It's also unclear how onroad diesel NOx d15N values were chosen and what uncertainties are. 4. First, if Figure 9 is the result of previous publication, it should be noted and cited. Second, Walters et al (2018) measured d15N of NO2 instead of NOx. It is questionable to compare that results with the simulation. Third, the simulation only considers direct emission signatures, while the observation should be influenced by both direct emissions and atmospheric processing. This adds more confusion to this comparison.

Minor remarks: Page 4, line 18, 'distinctive differences in d15N' is questionable, as there are significant overlap among different sources (Figure 1). This need to be clarified and interpreted appropriately. Page 5, line 17, 'NOx chemical lifetime (∼1 day) ' is over generalized and the NOx lifetime really depends on time and location. Also it needs references here. Page 5, line 20, how would only compare the emission sources be practical without considering other two factors. Table 1, the authors need rationale why the d15N of NOx values were selected from only the listed works rather than including others. Page 8, lines 5-8, the results obtained by Felix&Elliott were d15N-NO2 with passive sampler, but the authors compare the values with those obtained from dynamic flux chamber. The latter mainly measured NO directly emitted from soils. The authors should clarify this point and make proper selection of data as model input. Page 8, line 21, the authors need to clarify "these studies". The statement here is contradict to Table 1, which show the soils d15-NOx is adopted from one study Felix&Elliott 2014. Page 13 Figure 2, what year does the simulation show? page 5, line 33, why did the authors used NEI 2002 instead of the most recent one?

---

## Author Comment (AC1) · 19 Apr 2021

The authors adopted a mean d15N value for each source as the model input. However, large uncertainties are associated with each of the sources. The range of d15N values, not uncertainty, for any source is generally a function of equilibrium, kinetics, or reaction

progress happening in that source process. For example, automobiles show a wide range of both NOx amount and d15N values going from cold start to normal driving, but once the catalytic converter is warm the values are relatively constant because the NOx reduction by the CC becomes constant. We are using the average to account for these effects and for simplicity. In future work hope to explicating model the sources variation in SMOKE or land surface models, but that is well beyond the scope of this work. Fig. S10 shows the uncertainties of d15N values within the research area. For most of the grids, the uncertainties are less than 5 ‰ which is well below the difference in d15N values between any two of the emission sources. For those regions dominated by biogenic source, the uncertainties in d15N values are less than 10 ‰ which is also significantly below the difference in d15N values between the emission from biogenic source and all the other sources.

Page 8, lines 40-43, reasons were given here about why the results from passive sampling were excluded. What are the reasons why the passive sampling results were adopted for soil emissions? Most passive samplers are located near a particular source (such as near road) but sample the atmosphere not directly the source. Two problems arise in this situation, first is there will be other NOx sources besides the proximal one thus it is unclear what fraction of the NOx comes from the proximal source. Second, is that photochemical conversion and isotope exchange will alter source NO soon as it is converted into NO2 (usually what is sampled) or into other higher nitrogen oxides.

Page 9, lines 20-31, I am confused about how the d15N-NOx of gasoline values were chosen, as the authors show two different ones -2.5±1.5‰ and -2.7±1.8‰ and also show an equation in line 32. It's also unclear how on-road diesel NOx d15N values were chosen and what uncertainties are. The value -2.5±1.5‰ is the d15N of the NOx emission from on-road gas car with SCR equipped. The equation below were what we used to determine the d15N of NOx emission from on-road gas car. It is an equation of vehicle travel time. The vehicle travel time within each region with the same zip code were used for the calculation. The value -2.7±1.8‰ is the calculation result.

First, if Figure 9 is the result of previous publication, it should be noted and cited. Second, Walters et al (2018) measured d15N of NO2 instead of NOx. It is questionable to compare that results with the simulation. Third, the simulation only considers direct emission signatures, while the observation should be influenced by both direct emissions and atmospheric processing. This adds more confusion to this comparison. First, Fig. 8 (I think you may refer to Fig. 8 instead of Fig. 9, but the statement also applies to Fig, 9) is not from any previous publication, it is original of this manuscript. Second, Walters et al (2018) did include the d15N(NOx) values based on the measurements in Table 1 and 3. In order to be clear, the corresponding sentences was changed The first comparison is to the only direct measurements within the domain, which occurred in West Lafayette, IN. The $\delta$15N(NOx) values were inferred from the measured $\delta$15N(NO2) and the calculated $\delta$15N(NO2) shift (Walters, Fang, & Michalski, 2018). Third, we are aware of the atmospheric processing, as well as the chemical mechanism, while in this paper, we only focus on the effects from different emission sources. It is a series of works, after we gradually consider more and more factors, the simulated d15N values will be closer and closer to the corresponding measurements.

Page 4, line 18, 'distinctive differences in d15N' is questionable, as there are significant overlap among different sources (Figure 1). This need to be clarified and interpreted appropriately. The purpose of Fig. 1 is to show the range of d15N value for each NOx emission source, and the value we finally chose for this research. In sections 2.1-2.4, we explain how we determine the exact value we use of each source for the simulation. Once the value is determined, instead of a range, for each source, the concern of "overlapping" is dissolved.

Page 5, line 17, 'NOx chemical lifetime (âĹij1 day) ' is over generalized and the NOx lifetime really depends on time and location. Also, it needs references here. Yes, ∼1 day is not precise here. Thus, we improved the sentence as the following Secondly, atmospheric processes that mix the NOx emissions, blurring multiple emission sources within a mixing lifetime relative to the NOx chemical lifetime (2-7 hours), which depends

on its concentration and the occurring dominant chemistry, varied by time and locations (Laughner & Cohen, 2019). Nevertheless, the purpose of this sentence is to raise the question that the atmospheric process will lead to the mixture of the NOx emission during the NOx chemical lifetime.

Page 5, line 20, how would only compare the emission sources be practical without considering other two factors. Table 1, the authors need rationale why the d15N of NOx values were selected from only the listed works rather than including others. Because we need to control variables when doing the exploration. In order to explore all the factors, which determine the d15N of atmospheric NOx, we need to work on them one by one. Thus, this paper, we only focus on the first factor, which is the emission source.

Page 8, lines 5-8, the results obtained by Felix & Elliott were d15N-NO2 with passive sampler, but the authors compare the values with those obtained from dynamic flux chamber. The latter mainly measured NO directly emitted from soils. The authors should clarify this point and make proper selection of data as model input. Most passive samplers are located near a particular source (such as near road) but sample the atmosphere not directly the source. Two problems arise in this situation, first is there will be other NOx sources besides the proximal one thus it is unclear what fraction of the NOx comes from the proximal source. Second, is that photochemical conversion and isotope exchange will alter source NO soon as it is converted into NO2 (usually what is sampled) or into other higher nitrogen oxides. The dynamic chamber study is the closest to a direct measurement (similar to Walters et al. tail pipe collections) for soil emissions. It minimizes mixing in of other sources and converts all NO into NO2 rather than partitioning them between the two.

Page 8, line 21, the authors need to clarify "these studies". The statement here is contradict to Table 1, which show the soils d15-NOx is adopted from one study Felix & Elliott 2014. We looked into the d15N value of every sample in each cited research, took the overall average. This average value is very closed (less than 1 ‰ to the values from Felix & Elliott 2014. Sorry for being unclear in the manuscript, the sentences have

been adjusted in the updated version. Based on these studies we adopted a $\delta 15N$ value for NOx emissions from the soil of -34.3 ‰ which is closest to the average value of these previous studies (Li & Wang, 2008; Felix & Elliott, 2014; Yu & Elliott, 2017; Miller et al., 2018), to determine the emission rate of 15NOx from biogenic sources using Eq. (2) and (3).

Page 13 Figure 2, what year does the simulation show? page 5, line 33, why did the authors used NEI 2002 instead of the most recent one? The year for the simulation on Fig. 2 is 2002. As mentioned in the manuscript, at page 6. We used NEI 2002 because it is relevant to the same timeframe as the d15N measurements of the NOy we have. The year of d15N measurement at NADP sites within the Midwest, taken by our lab, is 2001-03. The measurements from other research group, published on peer-reviewed journal previously, are taken between late 1990s to the year 2009. Thus, NEI 2002 is more relevant than the most recent one.

The authors did not consider the uncertainties associated with emission inventories and the d15N signatures in their simulation. For example, the NEI inventories are known to contain large errors, especially for biogenic sources. As pointed out by the authors, the d15N signatures are also highly uncertain and span large ranges for individual sources. However, only an average value was used for each source. What if the d15N source signatures varied over space and time? Would this variability in the source signatures change significantly the simulated spatial and temporal patterns? Figure S10 shows the uncertainties of d15N values within the research area. For most of the grids, the uncertainties are less than 5 ‰ which is well below the difference in d15N values between any two of the emission sources. For those regions dominated by biogenic source, the uncertainties in d15N values are less than 10 ‰ which is also significantly below the difference in d15N values between the emission from biogenic source and all the other sources.

Regarding the plant canopy effect on biogenic NOx removal. As mentioned by the authors, soil-emitted NOx can be removed by overlying canopies to a large extent (up

to 75%). However, this effect was not considered in the simulation. I am curious to see if the simulated d15N patterns would be changed by explicitly considering this canopy effect. The purpose of this series of manuscripts is to set up an effective method to evaluate the accuracy of any given emission inventory. The role of vegetation in NOx removal by overlaying canopies, may or may not be included in an emission inventory. Our goal is to use the isotope signature to verify whether the assumption of plant canopy effect in the given emission inventory we test are appropriate.

All GMD articles of type "development and technical paper" require the provision of the code used. Therefore, it is required, that you add a Code Availability Section to your article, stating how to access the exact SMOKE and CMAQ code versions used for this publication. The GitHub repositories for the versions of SMOKE and CMAQ used for this research have been added into the Code Availability section.

Please also note the supplement to this comment:
https://gmd.copernicus.org/preprints/gmd-2020-322/gmd-2020-322-AC1-supplement.pdf